# Multiscale effective connectivity analysis of brain activity using neural ordinary differential equations

Yin-Jui Chang[1], Yuan-I Chen[1], Hannah M. Stealey[1], Yi Zhao[1], Hung-Yun Lu[1], Enrique Contreras-Hernandez[1], Megan N. Baker[1], Edward Castillo[1], Hsin-Chih Yeh[1,2], Samantha R. Santacruz[1,3,4]*

1 Biomedical Engineering, University of Texas at Austin, Austin, TX, United States of America, 2 Texas Materials Institute, University of Texas at Austin, Austin, TX, United States of America, 3 Electrical & Computer Engineering, University of Texas at Austin, Austin, TX, United States of America, 4 Institute for Neuroscience, University of Texas at Austin, Austin, TX, United States of America

* s.santacruz@austin.utexas.edu

## Abstract

Neural mechanisms and underlying directionality of signaling among brain regions depend on neural dynamics spanning multiple spatiotemporal scales of population activity. Despite recent advances in multimodal measurements of brain activity, there is no broadly accepted multiscale dynamical models for the collective activity represented in neural signals. Here we introduce a neurobiological-driven deep learning model, termed multiscale neural dynamics neural ordinary differential equation (msDyNODE), to describe multiscale brain communications governing cognition and behavior. We demonstrate that msDyNODE successfully captures multiscale activity using both simulations and electrophysiological experiments. The msDyNODE-derived causal interactions between recording locations and scales not only aligned well with the abstraction of the hierarchical neuroanatomy of the mammalian central nervous system but also exhibited behavioral dependences. This work offers a new approach for mechanistic multiscale studies of neural processes.

**Data Availability Statement:** All analyses were implemented using custom Python code. Code and data to replicate the main results is available at https://github.com/santacruzlab/msDyNODE.

## Introduction

The brain is a complex system exhibiting computational structure involving multiple spatial scales (from molecules to whole brain) and temporal scales (from submilliseconds to the entire lifespan) [1]. Effective connectivity (EC) is a type of brain connectivity that characterizes relationships between brain regions [2]. Unlike structural connectivity for anatomical links and functional connectivity for statistical dependencies, EC refers to a pattern of causal interactions between distinct areas. Multiscale effective connectivity (msEC) among brain regions provides essential information about human cognition [3] and behaviors such as motor preparation [4], motor adaptation [5], motor timing [6], decision making [7], and working memory [8]. To date, much research has primarily focused on extracting EC from a single modality of neural measurements (e.g., electrophysiology, functional magnetic resonance imaging, and [18]F-

**Funding:** This work was supported by the National Science Foundation (Award No. 2145412, SRS), the Cockrell School of Engineering at the University of Texas at Austin (Start-up funds, SRS), the National Institutes of Health (Award No. DA060543, HCY), and the National Science Foundation (Award No. 2404334, HCY). The funders had no role in study design, data collection and analysis, decision to publish, or preparation of the manuscript.

**Competing interests:** The authors have declared that no competing interests exist.

fludeoxyglucose positron emission tomography [3]) and typically makes simplifying assumptions in which neural dynamics are linear [9] or log-linear [10]. However, the lack of the integration between multiple modalities and the reality of nonlinear neural dynamics prevents us from uncovering a deeper and more comprehensive understanding of system-level mechanisms of motor behavior [11, 12].

msEC can be divided into within-scale and cross-scale EC, where the former indicates the causal interactions between neural elements at the same spatial and temporal scales and the latter specifies the causal interactions between neural elements at different spatial or temporal scales. Previous work has largely focused on inferring within-scale EC *via* multivariate autoregressive models [13], vector autoregressive models [14], psycho-physiological interactions [15], structural equation modeling [16–19], or dynamic causal modeling [20]. Despite emergence of the cross-scale analyses such as source localization [21] and cross-level coupling (CLC) [22], the fidelity of experimental implementation of source localization is limited and only the statistical dependencies are quantified by CLC. To reveal the directed interactions across spatiotemporal scales of brain activity, recent work has developed the generalized linear model-based multi-scale method [23]. However, experimental data indicate that local brain dynamics rely on nonlinear phenomena [24]. Nonlinear models may be required to generate the rich temporal behavior matching that of the measured data [25]. Taking the nature of nonlinearity in brain computations, we have previously proposed the NBGNet, a sparsely-connected recurrent neural network (RNN) where the sparsity is based on the electrophysiological relationships between modalities, to capture cross-scale EC [26]. Despite the success of capturing complex dynamics using a nonlinear model, we still lack an integrative method that can infer nonlinear msEC.

To analyze multiscale neural activity in an integrative manner, we introduce a multiscale modeling framework termed msDyNODE (multiscale neural dynamics neural ordinary differential equation). Neural ordinary differential equation (NODE) is a new family of deep neural networks that naturally models the continuously-defined dynamics [27]. In our method, within-scale dynamics are determined based on neurobiological models at each scale, and cross-scale dynamics are added as the connections between latent states at disparate scales (**Fig 1**). Using both simulation and an experimental dataset, we demonstrate that msDyNODE not only reconstructs well the multi-scale data, even for the perturbation tasks, but also uncovers multi-scale causal interactions driving cognitive behavior.

## Results

### Validation of msDyNODE framework using simulated Lorenz attractor

Since the Lorenz attractor model is a standard nonlinear dynamical system in the field with its simplicity and straightforward state space visualization [28, 29], we first demonstrate the msDyNODE framework using the simulated Lorenz attractor dataset. A Python program is employed to generate synthetic stochastic neuronal firing rates and local field potentials from deterministic nonlinear system. Two sets of Lorenz attractor systems are implemented to simulate activity at two scales: one to simulate firing rates at the single-neuron scale and another to stimulate local field potentials (LFPs) at the neuronal population scale. Without causal interactions between scales, the msDyNODE reconstructs well the Lorenz attractor parameters, simulated firing rates and LFPs (mean absolute error = 0.64 Hz for firing rate; = 0.18 μV for LFPs; **Fig 2A**). To evaluate the performance of the msDyNODE in the multiscale system, we mimic cross-scale interactions by adding causal connections between latent states of the two systems (**Fig 2B**). Although the fitting accuracy is relatively poorer than the systems without causal interactions (mean absolute error = 1.43 Hz for firing rate; = 2.58 μV for LFPs), the

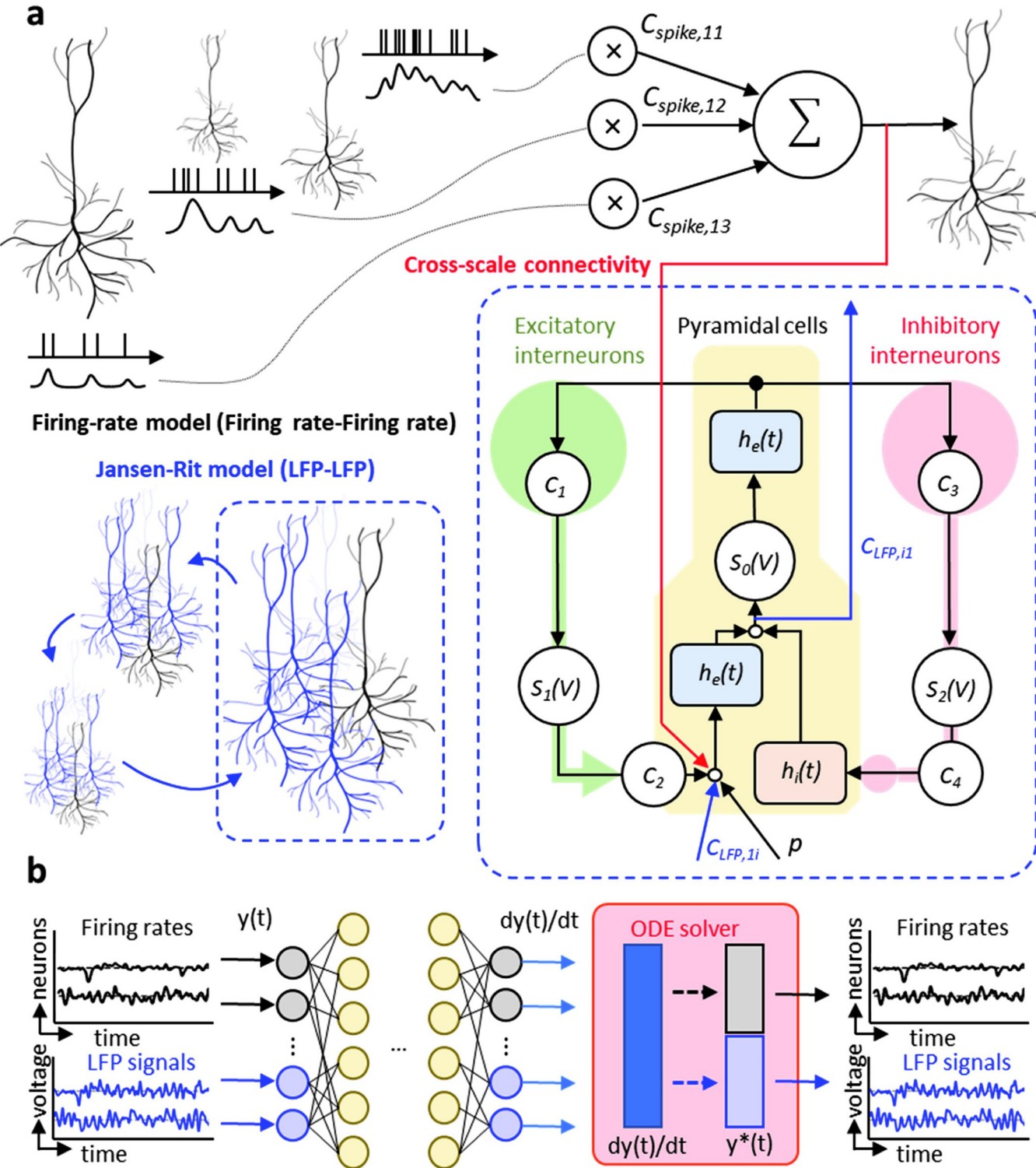

**Fig 1. The architecture of msDyNODE applied to multiscale LFP and firing rate.** (**a**) Firing rate-Firing rate model follows the firing-rate model. LFP-LFP model follows the Jansen-Rit model. Cross-scale connectivity between firing rates and LFPs is added between latent variables of two systems. (**b**) The schematics of msDyNODE for multiscale firing rate-LFP model.

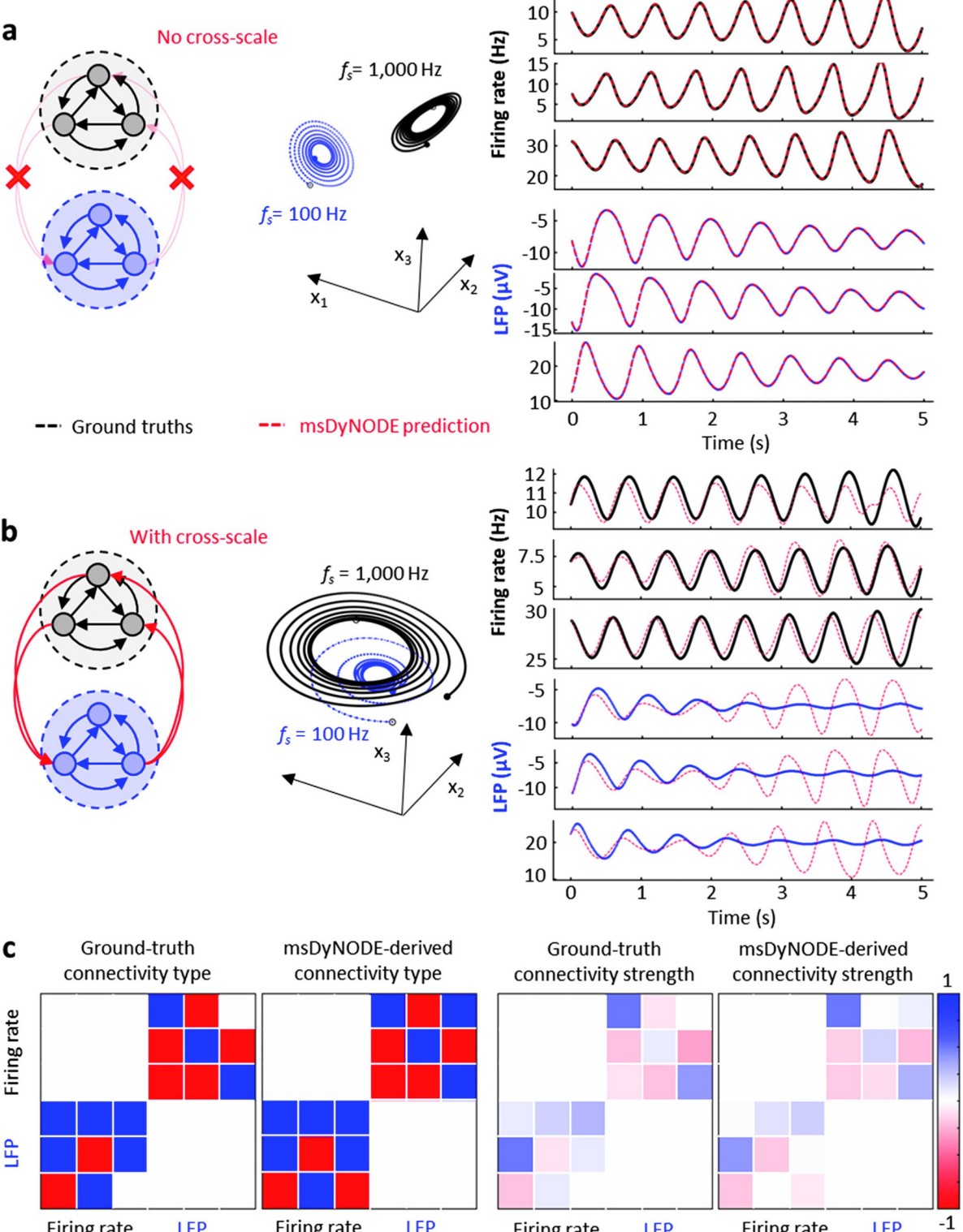

**Fig 2. msDyNODE applied to Lorenz attractor.** (**a**) The evolution of the Lorenz system in its 3-dimensional state space for firing rates (black) and LFPs (blue; left). The synthetic firing rates (black) and LFPs (blue), as well as the msDyNODE predictions (red dashed line), were plotted as a function of time (right). (**b**) The same as **a** but with cross-scale causal interactions. (**c**) Ground-truth and identified cross-scale communication types (left) and causal interactions (right) between synthetic firing rates and LFPs.

**Table 1. msDyNODE captures the Lorenz attractor parameters.** The predictions are summarized from 10 repeats of model training individually.

| Model parameters | Ground truth | Predictions (n = 10) |
|:---:|:---:|:---:|
| $\sigma_1$ | 10 | $10.09 \pm 0.04$ |
| $\rho_1$ | 28 | $28.02 \pm 0.06$ |
| $\beta_1$ | 2.67 | $2.69 \pm 0.03$ |
| $\sigma_2$ | 8 | $7.87 \pm 0.08$ |
| $\rho_2$ | 20 | $19.82 \pm 0.06$ |
| $\beta_2$ | 3.33 | $3.45 \pm 0.03$ |

msDyNODE still captures the signals and the Lorenz attractor parameters (**Table 1**). Notably, with the cross-scale interactions between systems, the msDyNODE can reconstruct the ground truth accurately for 2.5 seconds. Furthermore, we assess if the msDyNODE can identify the types (excitatory or inhibitory) and the strength of causal interactions (**Fig 2C**). Positive and negative causal strengths correspond to excitatory and inhibitory effects, respectively. The positive causality identified by the msDyNODE is true positive when the ground truth is also positive. It became a false positive if the ground truth is negative. The identification accuracy is 77 ±6% (**Fig 2C** left). We also find that msDyNODE successfully captures the cross-scale causal interactions (mean absolute difference between the ground-truth and estimated causality = 0.07; **Fig 2C** right). These simulations verify that msDyNODE is a reliable framework for modeling multiscale systems.

## msDyNODE outputs reconstruct well experimentally-acquired firing rate and field potential signals

Firing rate and LFP activity are simultaneously recorded in the left dorsal premotor (PMd) and primary motor cortex (M1) of rhesus macaques (N = 2) while performing a center-out brain-machine interface (BMI) task [30–34] (**Fig 3**; see **Materials and Methods**). Multi-scale firing rate and LFP are acquired with the same set of electrodes but undergoing different pre-processing procedures (**Fig 3A**). During the center-out BMI task, the subjects volitionally modulate brain activity to move the cursor from the center to one of the eight peripheral targets. When BMI perturbation task is implemented, the subjects need to reassociate the existing neural patterns with the new direction [32, 35]. The increasing deviation shown in our simulation (**Fig 2**) is not the problem in our case with the average trial less than 2.5 seconds. The msDyNODE for the firing rate-LFP modeling is developed based on rate model [36–38] and Jansen-Rit model [39] (**Fig 1**; see **Materials and Methods**). By fitting the msDyNODE to the experimental datasets, we demonstrate the goodness-of-fit of the proposed multiscale framework in modeling multiscale brain activity using correlation and mean absolute error metrics (**Fig 4**). Correlation between ground truth data and the msDyNODE-predicted data defines a linear relationship between the real and predicted signal, with a strong correlation ($> 0.7$) indicating consistent temporal co-variation between the two data up to a constant amplitude scaling. Mean absolute error (MAE), on the other hand, measures error in signal amplitude timepoint by timepoint but without describing the overall relationship between the data. Together, high correlation and low MAE indicate that the data co-vary together and any scaling difference between the real and predicted data is small. We find that indeed there is high correlation between ground truth data and msDyNODE-predictions, with msDyNODE primarily capturing the LFP activity below 30 Hz (**Fig 4A**). This observation is consistent with the fact that LFP neural dynamics are dominated by lower frequencies. Therefore, for the rest of

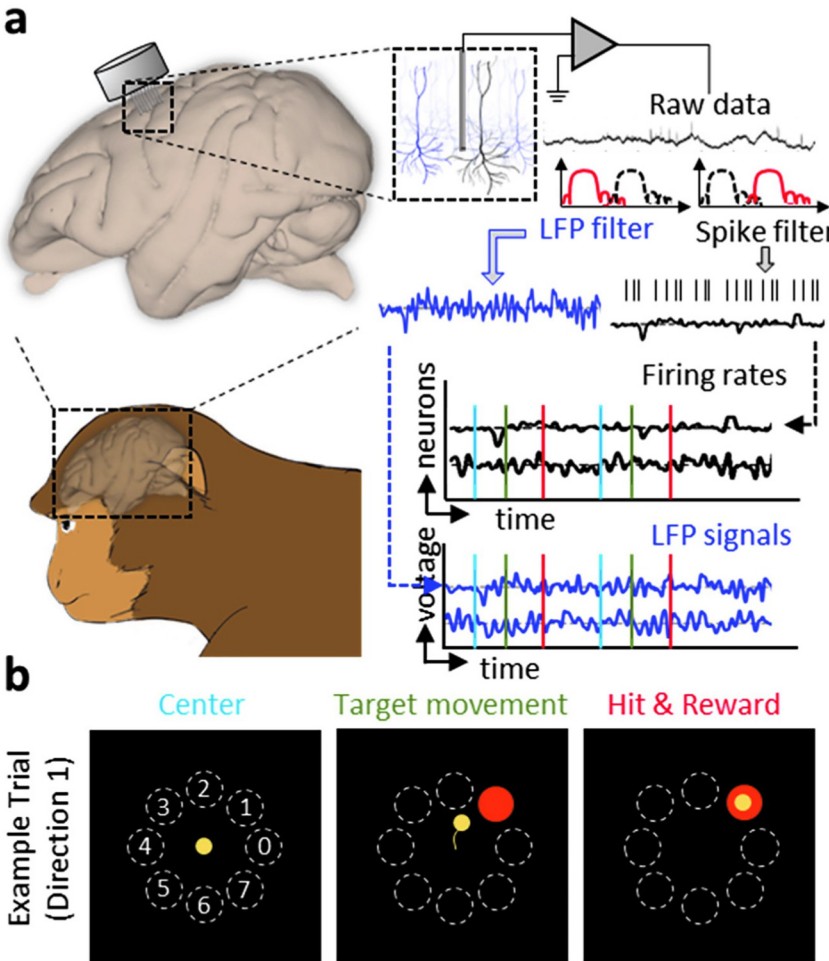

**Fig 3. Data acquisition and experimental task design for multiscale neural signals.** (**a**) Simultaneous recording of firing rates and LFP signals. (**b**) The visual feedback task contains eight different cursor movements, each corresponding to one of the eight outer targets. The color-coded tasks are also indicated in **a**.

the evaluations, we focus on the performance in the frequency range of 0 and 30 Hz. Overall, the msDyNODE well reconstructed the firing rates (median of MAE = 0.74 Hz) and LFPs (median MAE = 24.23 μV; **Fig 4B**). In addition, we find that the performance of the msDy-NODE is target direction-independent, with a similar MAE over eight target directions for both firing rates and LFPs (**Fig 4C**). Interestingly, the reconstruction performances for firing rates and LFPs are not independent (**Fig 4D**). Good performance on certain channels indicated similarly good performance for different signal types, and vice versa. Surprisingly, the modeling performance for firing rates remains high over hundreds of trials even when a perturbation is introduced to increase the task difficulty (**Fig 4E**). However, the modeling performance for LFP gradually improves over trials, which may indicate that LFP dynamics become more predictable. Furthermore, the performance holds when applying the msDyNODE to a different monkey dataset (i.e., that it is not trained on), indicating that the msDyNODE is generalizable across different sessions and subjects (**Fig 5**). With a larger number of spiking units and LFPs recorded in this subject, it is expected that the msDyNODE can reconstruct LFP more accurately. The only difference in the reconstruction performance is that the firing rate predictions were worse during the first half of the experimental sessions, followed by

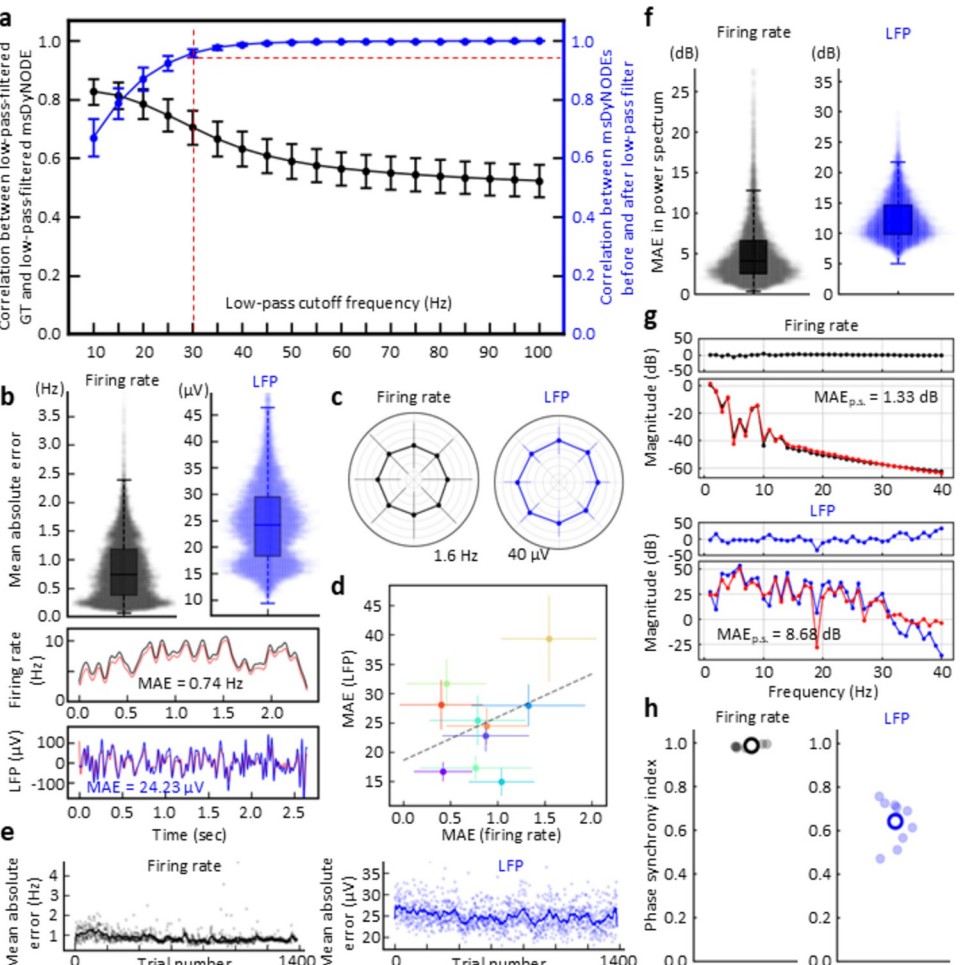

**Fig 4. msDyNODE captures and reconstructs the latent dynamics in the center-out BMI task for Monkey A.** (**a**) Correlation coefficient between ground truth (GT) signals and msDyNODE predictions (black) as a function of low-pass cutoff frequency (error bars, s.t.d.). In addition, we show correlation between msDyNODE before and after low-pass filter (blue). (**b**) Boxplots and swarmplots of the mean absolute errors in firing rates and LFPs (top). The representative GT and msDyNODE with the MAE equaling to the median values of all the MAEs (bottom). (**c**) Error bars of the MAE over eight different target directions presented in polar coordinates (error bars, s.t.d.). (**d**) Scatter plots of the MAE over recording channels (error bars, s.t.d.). (**e**) MAE values of firing rates and LFPs over trials. Dim points represent average MAE (n = 10) at each trial. (**f**) Boxplots and swarmplots of the mean absolute errors in power spectrum for firing rates and LFPs. (**g**) The representative power spectrum from GT and msDyNODE with the selected example in **Fig 4B**. (**h**) Scatter plots of PSI values for firing rates and LFPs. Empty circles indicate overall average PSI values. Dim points represent average PSI over trials for each recording channel.

increasing accuracy for the second half of the recording sessions (**Fig 5E**). This may indicate the neural dynamics were less stable during the first half of the sessions and thus more challenging to be captured. Beyond MAE in the time domain, we also assess MAE in the frequency domain and phase synchronization in the phase domain (**Figs 4F–4H, 5F–5H**; see **Materials and Methods**). Overall, the msDyNODE captures the signal's power for both Monkey A (**Fig 4F and 4G**) and Monkey B (**Fig 5F and 5G**). Notably, phase synchronization is recognized as a fundamental neural mechanism that supports neural communication and plasticity [40]. Therefore, the model performance in the phase domain is crucial. We demonstrated that msDyNODE-predictions are in sync with ground truth by showing most of the predictions have a phase synchrony index greater than 0.5 (**Figs 4H and 5H**). These experimental results

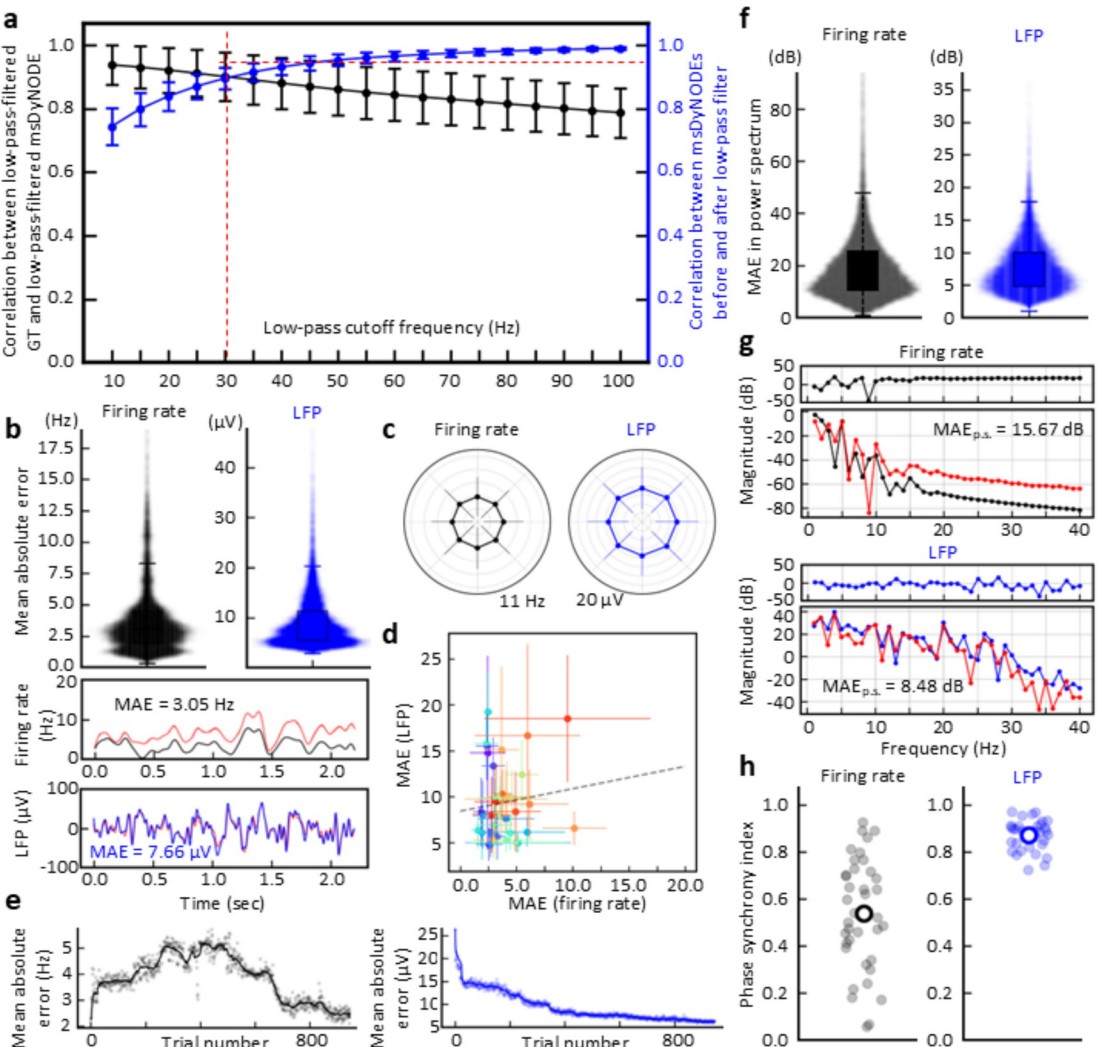

**Fig 5. msDyNODE captures and reconstructs the latent dynamics in the center-out BMI task for Monkey B.** (**a**) Correlation coefficient between ground truths (GT) and msDyNODE (black) and between msDyNODE before and after low-pass filter (blue) as a function of low-pass cutoff frequency (error bars, s.t.d.). (**b**) Boxplots and swarmplots of the mean absolute errors in firing rates and LFPs (top). The representative GT and msDyNODE with the MAE equaling to the median values of all the MAEs (bottom). (**c**) Error bars of the MAE over eight different target directions presented in polar coordinates (error bars, s.t.d.). (**d**) Scatter plots of the MAE over recording channels (error bars, s.t.d.). (**e**) MAE values of firing rates and LFPs over trials. Dim points represent average MAE (n = 38) at each trial. (**f**) Boxplots and swarmplots of the mean absolute errors in power spectrum for firing rates and LFPs. (**g**) The representative power spectrum from GT and msDyNODE with the selected example in **Fig 5B**. (**h**) Scatter plots of PSI values for firing rates and LFPs. Empty circles indicate overall average PSI values. Dim points represent average PSI over trials for each recording channel.

validated that msDyNODE can capture the dynamics hidden in the multiscale brain systems, and msDyNODE can be generalized to different sessions and different subjects.

## msDyNODE decodes underlying behavior *via* multiscale effective connectivity

In msDyNODE, the msEC can be derived from the parameters that indicate the causal influence that the latent states of a neural system exert over those of another system. The average connectivity for each target direction is calculated by subtracting the grand-averaged

connectivity from the average connectivity within each target (**Fig 6A**). For each direction, the bi-directional msEC is divided into two parts (upper and lower triangular connectivity matrix) and visualized respectively (**Fig 6B**). Most of the msEC remained similar across target directions, indicating the common patterns of voluntary movement. To investigate if there existed unique patterns of excitatory and inhibitory subnetworks across directions, we quantified the individual subnetworks using common graph properties such as number of edges, average clustering, and total triangles (**Fig 6C**). Interestingly, these graph properties are different across the eight target directions, revealing the excitatory and inhibitory neural dynamics exhibited unique connectivity patterns relating to target direction. Thus, msDyNODE is demonstrated to be capable of capturing the multiscale effective connectivity patterns underlying behaviors.

## Discussion

Large populations of individual neurons coordinate their activity together to achieve a specific cognitive task, highlighting the importance of studying the coordination of neural activity. Over the past decades, we have learned much about the human cognitive behaviors and viewed an explosive growth in the understanding of single neurons and synapses [41, 42]. However, we still lack a fundamental understanding of multiscale interactions. For decades a critical barrier to multiscale study was the recording technologies available, forcing scientists to choose either the microscale or macroscale, with few researchers addressing on the interactions between scales. Neurophysiologists, for example, often focused on single-neuronal activity to investigate the sensory consequences of motor commands with the bottom-up approach [43], without the consideration of brain rhythm. Instead, cognitive neuroscientists pay attention to the neural oscillations at a larger scale (e.g., electroencephalography) with the top-down approach to establish the links between brain rhythm and cognitive behaviors [44], disregarding the spiking activity of single neurons. With the advancement of multi-modal measurements, there is an unmet need for an integrative framework to analyze multiscale systems. In the present study, we propose msDyNODE to model the multiscale signals of firing rates and field potentials, and then infer multiscale causal interactions that exhibit distinct patterns for different motor behaviors.

To the best of our knowledge, this is the first demonstration of a NODE applied to model multiscale neural activity. Assuming brain computation as a nonlinear operator [45–51], we employ a deep learning technique to approximate the nonlinear mapping of the state variables in dynamic systems. Different deep learning architectures are tailored for specific tasks. Common examples include convolution neural networks for image recognition [52], recurrent neural networks (RNN) for sequential data [53], transformers for natural language processing tasks [54], and generative adversarial networks for generating authentic new data [55] and denoising [56]. While RNNs are a powerful approach to solve the dynamic equations [57, 58], it may fail to capture faster dynamics information or introduce artifacts by matching the sampling rates between signals. In contrast to the RNN which describes the complicated state transformation at discretized steps for time-series inference, the proposed msDyNODE models continuous dynamics by learning the derivative of the state variables [27], indicating that both slow and fast dynamics can be captured. Such a capability is crucial for multiscale modeling since the system dynamics vary at different scales. Additionally, NODE allows us to define the multiscale system by customizing the differential equations in the network, in which we can investigate the physiological interpretation of the modeled systems. It is worth noting that the nonconstant sampling can be addressed by preprocessing the NODE output with the observation mask [59]. Therefore, unmatched sampling rates between modalities can be resolved by feeding individual observation masks, respectively. Furthermore, in the real world,

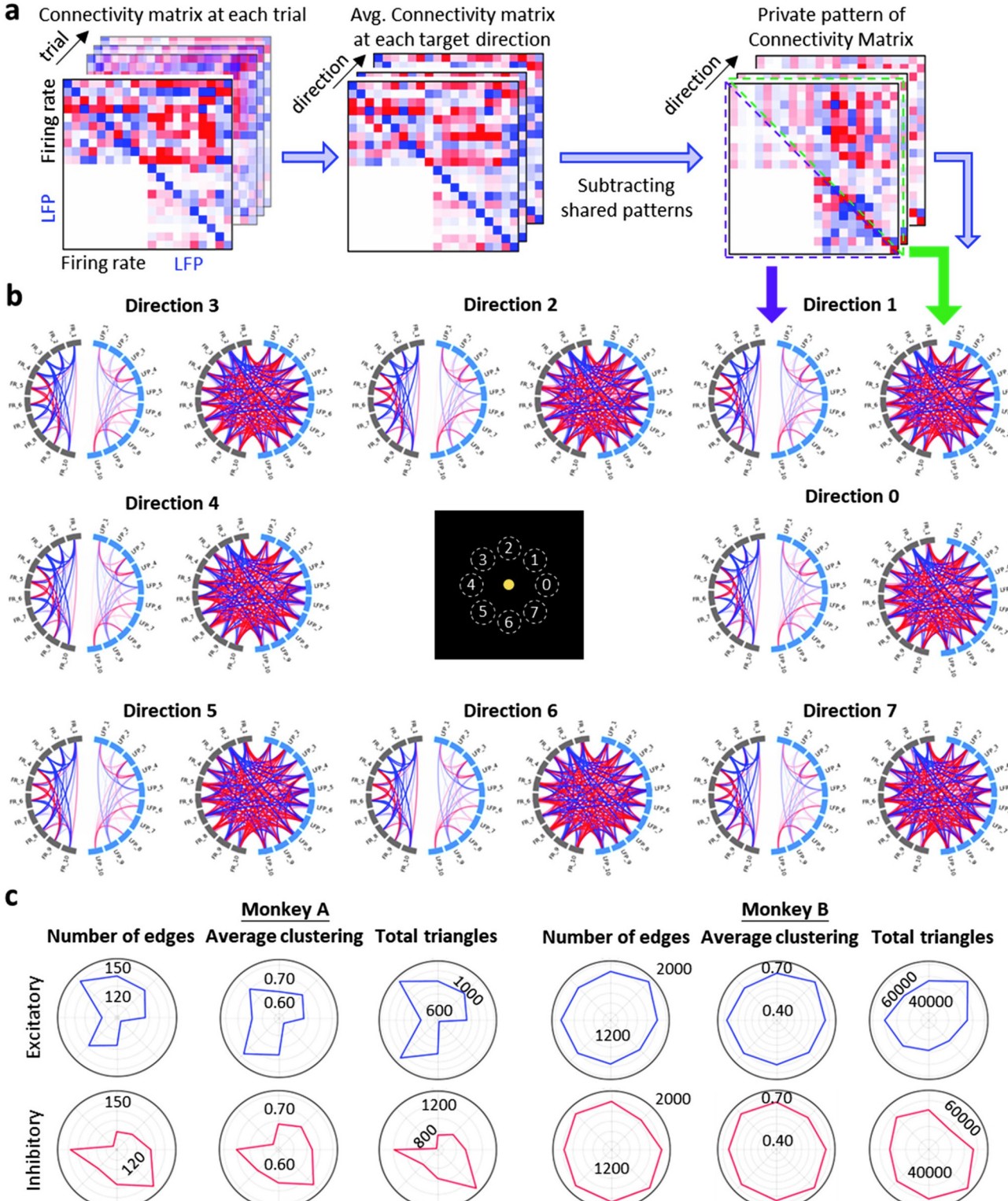

**Fig 6. msDyNODE captures msEC patterns underlying behaviors.** (**a**) Workflow to obtain the private pattern of connectivity matrix for each target direction from msDyNODE-inferred msEC. (**b**) Circular connectivity graphs of lower (left) and upper (right) triangular msEC matrix for each target direction. (**c**) Graph properties (number of edges, average clustering, number of total triangles) over eight different target directions presented in polar coordinates for Monkey A and B, and excitatory and inhibitory subnetworks, respectively.

not all the signals can be measured at fixed time intervals. The missing data issue can thus introduce artefacts using a conventional approach which assumes the signals are sampled regularly. While there exists several methods, such as dropping variables, last observation carried forward and next observation carried backward, linear interpolation, linear regression, or imputation dealing with missing data [60], none of them serves as good way to deal with this issue because they add no new information but only increase the sample size and lead to an underestimate of the errors. The proposed framework also holds great potential to be an alternative approach dealing with missing data commonly seen in the real world.

Comparing with existing biophysical models of brain functioning, including NetPyNE [61], modified spectral graph theory model (M-SGM [62]), and SGM integrated with simulation-based inference for Bayesian inference (SBI-SGM [63]), we demonstrate that msDyNODE is superior these approaches. msDyNODE showed smaller MAEs in both time and frequency domains, and greater phase synchronization with the ground truth signals (**S1 Fig**). The potential reason for relatively poor performance in NetPyNE may be due to inaccurate modeling. Indeed, NetPyNE is a powerful tool to define the model at molecular, cellular, and circuit scales when the model parameters such as populations, cell properties, and connectivity are accurate. Although NetPyNE also provides evolutionary algorithms beyond the grid parameter search to perform parameter optimization and exploration, the improper selection of parameters and their ranges to be optimized can degrade the performance. Furthermore, msDyNODE exhibits better performance than both versions of SGMs (**S2 Fig**). The rationale for why msDyNODE models the real multiscale brain signals better than SGMs may be due to the consideration of nonlinear brain dynamics and spatially varying parameters. Another advantage of msDyNODE over NetPyNE, M-SGM, and SBI-SGM is the adaptability of the new model. For msDyNODE, the user can easily modify the differential equation sections in the script. However, NetPyNE requires the development of an external module in NEURON [64, 65]. For M-SGM and SBI-SGM, the new transfer function is required to derive from the new or customized models.

While msDyNODE provides accurate analysis for multiscale systems, its cost lies in the optimal selections of neural models. At the scale of firing rate, integrate-and-fire model and its variants (leaky integrate-and-fire [66, 67] and quadratic integrate-and-fire [68, 69]) are all plausible options. At the scale of field potential activity, the candidate model includes Jansen-Rit model [39] that characterizes three populations (pyramidal cells, excitatory interneurons, and inhibitory interneurons) and Wilson-Cowan model [70] that refers to two coupled populations (excitatory and inhibitory). Suboptimal selections of neural models may result in misleading conclusions. To avoid suboptimal model selection, probabilistic statistical measures such as Akaike Information Criterion [71, 72], Bayesian Information Criterion [73], and minimum description length [74, 75] can be implemented to ensure the correct selection of the neural models. Furthermore, hours of network training time are another issue for quick implementation. In future work, transfer learning [76] from the previously trained network may be a possible strategy to improve computation time by potentially speeding up the convergence of the learning process.

Recent evidence suggests that signal changes on multiple timescales at multiple levels in the motor system allow arbitration between exploration and exploitation to achieve a goal [77–80]. Still, the role of cross-scale, as well as within-scale, causal interactions in motor learning remains incompletely understood [78, 79, 81]. In this work, we utilize the msDyNODE to study the essential brain function that modulates the motor commands to achieve desired actions, showing distinct dynamic patterns underlying different behaviors. Although existing estimators of causal brain connectivity (e.g., Granger causality [82] and directed transfer function (DTF [83])) provide disparate graph properties (**S3 and S4 Figs**), Granger causality

supports our observation that both excitatory and inhibitory msEC exhibit unique patterns relating to target directions. In contrast, DTF failed to demonstrate unique patterns, which may be due to its inability to be divided into excitatory and inhibitory subnetworks. While both existing estimators are powerful tools for characterizing functional coupling between brain regions, they primarily reflect the patterns of statistical dependence. To better reveal the causal interactions that align with the actual mechanisms of brain function, it is suggested to assess effective connectivity using a mechanistic model, such as msDyNODE. Taken together, our work represents an important step forward towards multiscale modeling of brain networks for mechanistic understanding of neural signaling. The underlying multiscale dynamics embedded in msDyNODE illustrate how the individual neurons and populations of neurons communicate across scales, which is a key factor in uncovering the mechanisms of brain computations and the mediation of the behaviors.

## Materials and methods

### Ethics statement

All the experiments were performed in compliance with the regulation of the Institutional Animal Care and Use Committee at the University of Texas at Austin.

### Experimental protocol

Two male rhesus macaques are used in the behavioral and electrophysiological experiments. Before the experimental session, we run a calibration session. During calibration, the subject passively observes a cursor moving from the center target toward a randomly generated peripheral target (in one of eight possible positions), followed by the cursor movement back to the center. In addition to providing continuous visual feedback, we also reinforce the behavior and neural activity by delivering a small juice reward directly into the subject's mouth. The neural data is recorded for approximately three and a half minutes (or reaching ~6 trials per target direction). A Kalman filter (KF) is employed as the decoder to map the spike count from each unit to a two-dimensional cursor control output signal [84, 85]. While the KF decodes both the intended position and velocity, only the velocity is used to estimate the position at the next time point based on kinematic equations of motion. To increase the initial performance and reduce directional bias, we conduct daily, 10-minute closed-loop decoder adaptation (CLDA) [85–89] sessions. Both the decoder and neural activity adapt to complete center-out tasks with consistent trial times and straight path lengths to each target. After the calibration session, the main task is manually initiated. The subject then completes a BMI task called "center-out" [90–92]. During the task, spiking activity is recorded online to produce cursor control commands in real-time. Spikes for each unit are summed over a window of 100 millisecond and serve as the input to the decoder. The neural activity is then transformed into a "neural command" by applying the dot product of the spike count vector to the Kalman gain matrix. Cursor position is updated iteratively by adding the cursor position to the product of velocity, which is determined by the neural command, times update time (100 ms). In each trial, the subjects control the velocity of a computer cursor to move from the center target toward one of eight outer targets. Only one peripheral target is presented on a given trial. The order of the appearance of the target is pseudorandomly selected; for every eight consecutive trials, each target is shown once in a random order. The 8 targets were radially distributed from 0˚ to 360˚ (0˚, 45˚, 90˚, 135˚, 180˚, 225˚, 270˚, 315˚) at equal distances from the center (10 centimeters). Upon successful completion of moving and holding the cursor at the peripheral target for 0.2 seconds, the target turns green (cue for success), and a small juice reward is dispensed directly into the subject's mouth. The cursor then automatically appears at the

center of the screen to initiate another new trial. Subjects can fail the task in two ways: (1) failure in holding the cursor at the center target or the peripheral target for 0.2 seconds or (2) failure in reaching the peripheral target within specified time (10 seconds). The subject has 10 chances to complete a successful trial before the task automatically moves onto the next target. During the BMI tasks, we also implement perturbation task by perturbing the decoder using a visuomotor rotation in which the cursor movements are rotated by an angle. The subjects then need to reassociate the existing neural patterns with new directions [32, 35].

## Spike trains and LFP data

The extracellular single and multi-unit activity in the left primary motor cortex (M1) and dorsal premotor cortex (PMd) are recorded using a 64- or 128-channel chronic array (Innovative Neurophysiology, Inc., Durham, NC; **Fig 3A**) from both subjects. The spike trains are acquired at 30 kHz sampling frequency, and the LFPs are acquired at 1 kHz sampling frequency. After excluding the recording channels that fail to capture activity (average firing rate < 1 Hz), 10 (Monkey A) and 38 (Monkey B) channels are considered for analysis. Cursor movements are tracked using the custom-built Python-based software suite. Neuronal signals are recorded using Trellis (Ripple Neuro, UT, USA) interfacing with Python (v3.6.5) via the Xipppy library (v1.2.1), amplified, digitized, and filtered with the Ripple Grapevine System (Ripple Neuro, UT, USA).

## Multiscale dynamics modeling with neurobiological constraints

We define a multi-scale dynamics network as a collection of neural recordings from different modalities (e.g., spike trains, LFPs, EEGs, fast-scan cyclic voltammetry, calcium imaging, functional magnetic resonance imaging, and functional near-infrared spectroscopy). A generic multi-scale dynamics system, where the evolution of latent variables and the output was described by the nonlinear functions of latent states and corresponding inputs, for $M$ modalities is as follows,

$$\dot{\mathbf{x}}_i = \sum_{j=1}^{M} f_{ij}(\mathbf{x}_j, \theta_j, t), \quad i \in [1, 2, \dots, M]$$

$$\mathbf{y}_i = g_{ii}(\mathbf{x}_i, \theta_i, t), \quad i \in [1, 2, \dots, M]$$

where $\mathbf{x}_i$, $\mathbf{y}_i$ represent the latent state variables and the observations for $i$th modality, respectively, $f_{ij}$ denotes within-scale ($i = j$) and cross-scale ($i \neq j$) dynamics parameterized by $\theta_j$, and $g_{ii}$ is the observation model in each modality. In this work, we focus on firing rates and LFPs (**Figs 1 and 3**), referred to as multi-scale signals. In addition, to enable the interpretability of the deep learning model, we introduce neurobiological constraints in our proposed network. Constraints including integration of modeling across different scales, the nature of the neuron model, regulation and control through interplay between excitatory and inhibitory neurons, and both local within- and global between-area connectivity have been reported to make neural network models more biologically plausible [93]. How are these neurobiological constraints implemented in the proposed approach are described in the following sections.

The multi-scale dynamics modeling for firing rate activity and LFP are based on well-established neurobiological models can be divided into three parts: (1) firing rate-firing rate within-scale model, (2) LFP-LFP within-scale model, and (3) firing rate-LFP cross-scale model. The rate model is employed as the firing rate-firing rate inference model with $N_{tol}$ coupled neurons

[36–38]:

$$\frac{dx_{FR,i}}{dt} = \frac{x_{FR,i} + \text{sigm}\left(\sum_{j=1}^{N_{tol}}\left(C_{hidden\ FR,ij}\frac{dx_{FR,j}}{dt}(t) + C_{FR,ij}x_{FR,j}(t)\right)\right)}{-\tau_m},$$

where $x_{FR,i}$ represents the membrane voltage of neuron $i$, $\tau_m$ denotes the membrane time constant, $C_{FR,ij}$ and $C_{hidden\ FR,ij}$ represents two types of causal interactions between presynaptic neuron $j$ and postsynaptic neuron $i$. For the LFP-LFP within-scale model, we implement the Jasen-Rit model to describe the local cortical circuit by second-order ODEs [39]:

$$\ddot{x}_{LFP,i0} = Aa\,\text{sigm}_0(x_{LFP,i1}(t) - x_{LFP,i2}(t)) - 2a\dot{x}_{LFP,i0} - a^2x_{LFP,i0}(t),$$

$$\ddot{x}_{LFP,i1} = Aa[p_i(t) + C_2\,\text{sigm}_1(C_1x_{LFP,i0}(t)) + p_{mu}] - 2a\dot{x}_{LFP,i1} - a^2x_{LFP,i1}(t),$$

$$\ddot{x}_{LFP,i2} = BbC_4\text{sigm}_2(C_3x_{LFP,i0}(t)) - 2b\dot{x}_{LFP,i2} - b^2x_{LFP,i2}(t),$$

$$p_i(t) = \sum_{j}^{N_{tol}}C_{LFP,ij}\,\text{sigm}_0(x_{LFP,j1}(t) - x_{LFP,j2}(t)),$$

where *sigm()* is a sigmoid function, $A$ and $B$ represent the maximum amplitude of the excitatory and inhibitory postsynaptic potentials (PSPs), $a$ and $b$ denote the reciprocal of the time constants of excitatory and inhibitory PSPs, $p_{mu}(t)$ represents the excitatory input noise of the neuron $i$, and $p(t)$ represents the excitatory input of the neuron $i$ from other neurons.

For the cross-scale model that identifies and quantifies cross-scale communications, we consider the causal interactions between the hidden states (membrane voltage of single neuron for spike; membrane potential of pyramidal, inhibitory, and excitatory neurons) as the effective connectivity:

$$h_{LFP} = Ch_{FR} + \varepsilon,$$

where $C$ represents the cross-scale causal interactions, and $\varepsilon$ denotes the error, which includes inputs from other units which are not explicitly considered. Note here that the cross-scale interactions are defined to be unidirectional and linear due to fact that the LFP are defined as the summed and synchronous electrical activity of the individual neurons. After implementing the cross-scale causal interactions as the excitatory input of the neurons, the second ordinary differential equation in the Jasen-Rit model becomes as follows,

$$\ddot{x}_{LFP,i1} = Aa[p_i(t) + C_2\,\text{sigm}_1(C_1x_{LFP,i0}(t)) + p_{mu} + C_{FR-LFP,ij}x_{FR,j}] - 2a\dot{x}_{LFP,i1} - a^2x_{LFP,i1}(t),$$

Taken together, combining the above equations, our multiscale dynamics model for spike and field potential can be written as follows, where $F_{FR-FR}$ and $F_{LFP-LFP}$ represent the within-scale dynamics equations while $F_{FR-LFP}$ denotes the cross-scale dynamics equations:

$$\frac{d\mathbf{x}}{dt} = \begin{bmatrix}\dfrac{d\mathbf{x}_{FR}}{dt}\\[2mm]\dfrac{d\mathbf{x}_{LFP}}{dt}\end{bmatrix} = \begin{bmatrix}F_{FR-FR} & \mathbf{0}\\ \mathbf{0} & F_{LFP-LFP}\end{bmatrix}\begin{bmatrix}\mathbf{x}_{FR}\\ \mathbf{x}_{LFP}\end{bmatrix} + \begin{bmatrix}\mathbf{0} & \mathbf{0}\\ F_{FR-LFP} & \mathbf{0}\end{bmatrix}\begin{bmatrix}\dfrac{d\mathbf{x}_{FR}}{dt}\\[2mm]\dfrac{d\mathbf{x}_{LFP}}{dt}\end{bmatrix} + \begin{bmatrix}\mathbf{b}_{FR}\\ \mathbf{0}\end{bmatrix}.$$

**Multiscale neural dynamics neural ordinary differential equation (msDyNODE).** Popular models such as recurrent neural networks and residual networks learn a complicated

transformation by applying a sequence of transformations to the hidden states [27]: $\mathbf{h}_{t+1} = \mathbf{h}_t + f(\mathbf{h}_t, \theta_t)$. Such iterative updates can be regarded as the discretization of a continuous transformation. In the case of infinitesimal update steps, the continuous dynamics of the hidden states can be parameterized with an ordinary differential equation (ODE):

$$\frac{d\mathbf{h}(t)}{dt} = f(\mathbf{h}(t), \theta, t).$$

A new family of deep neural networks, termed the neural ODE (NODE), was thus introduced to parameterize the $f$ using a neural network [27]. The output of the NODE was then computed using any differential equation solver (e.g., Euler, Runge-Kutta methods). In this work, we utilize Runge-Kutta method with a fixed time step of 1 ms. The resulting msDy-NODE model consists of 7 layers with 1,480 and 18,392 trainable parameters for Monkey A and B, respectively. NODE exhibits several benefits, including memory efficiency, adaptive computation, and the capability of incorporating data arriving at arbitrary times. Recent work proposed a NODE-based approach with a Bayesian update network to model the *sporadically* observed (i.e., irregular sampling) multi-dimensional time series dataset [59]. Therefore, NODE serves as powerful tool for multi-scale data analysis.

**Synthetic Lorenz attractor.** The Lorenz attractor is a simple but standard model of a nonlinear, chaotic dynamical system in the field [28, 94]. It consists of nonlinear equations for three dynamic variables. The state evolutions are derived as follows,

$$\dot{x}_1 = \sigma(x_2 - x_1)$$

$$\dot{x}_2 = x_1(\rho - x_3) - x_2$$

$$\dot{x}_3 = x_1 x_2 - \beta x_3$$

The standard parameters are $\sigma = 10$, $\rho = 28$, and $\beta = 8/3$. The Euler integration is used with $\Delta t = 0.001$ (i.e. 1 ms). We first simulate two sets of Lorenz attractor systems with different parameter sets ($\sigma_1 = 10$, $\rho_1 = 28$, $\beta_1 = 8/3$, $\sigma_2 = 8$, $\rho_2 = 20$, and $\beta_2 = 10/3$) but without cross-scale interactions:

$$\dot{x}_1 = \sigma_1(x_2 - x_1),$$

$$\dot{x}_2 = x_1(\rho_1 - x_3) - x_2,$$

$$\dot{x}_3 = x_1 x_2 - \beta_1 x_3,$$

$$\dot{x}_4 = \sigma_2(x_5 - x_4),$$

$$\dot{x}_5 = x_4(\rho_2 - x_6) - x_5,$$

$$\dot{x}_6 = x_4 x_5 - \beta_2 x_6,$$

with one system for a population of neurons with firing rates given by the Lorenz variables and another system for LFPs given by the Lorenz variables (**Fig 2**). We start the Lorenz system with a random initial state vector and run it for 6 seconds. We hypothesize that the neural activity consists of multiple marginally stable modes [95, 96]. The last five seconds were selected to ensure marginal stability in the simulation. Three different firing rates and LFPs were then generated with different sampling rates (1,000 Hz for spikes and 100 Hz for LFPs).

Models are trained with ten batches of 1-second data with randomly selected starting points for 1,000 iterations.

To evaluate the fitting performance of the msDyNODE with the Lorenz systems with cross-scale interactions, we then simulate two sets of Lorenz attractor systems with different parameter sets ($\sigma_1 = 8$, $\rho_1 = 28$, $\beta_1 = 8/3$, $\sigma_2 = 10$, $\rho_2 = 20$, and $\beta_2 = 10/3$) and cross-scale interactions:

$$\dot{x}_1 = \sigma_1(x_2 - x_1) + 0.1x_4 + 0.2x_5 + 0.3x_6,$$

$$\dot{x}_2 = x_1(\rho_1 - x_3) + 0.5x_4 - 0.1x_5 + 0.1x_6,$$

$$\dot{x}_3 = x_1x_2 - \beta_1x_3 - 0.2x_4 + 0.1x_5,$$

$$\dot{x}_4 = \sigma_2(x_5 - x_4) + 0.5x_1 - 0.1x_2,$$

$$\dot{x}_5 = x_4(\rho_2 - x_6) - x_5 - 0.2x_1 + 0.1x_2 - 0.3x_3,$$

$$\dot{x}_6 = x_4x_5 - \beta_2x_6 - 0.1x_1 - 0.2x_2 + 0.4x_3,$$

All the other simulation settings remain the same as above.

**Phase synchrony assessment.** We apply the Hilbert transform, $\mathbf{HT}[\cdot]$, on a pair of signals, $\mathbf{s}_1(t)$ and $\mathbf{s}_2(t)$, in order to obtain the analytical signals, $\mathbf{z}_1(t)$ and $\mathbf{z}_2(t)$.

$$z_i(t) = s_i(t) + j\mathbf{HT}[s_i(t)] = \mathbf{A}_i(t)e^{j\Phi_i(t)}$$

$$\mathbf{HT}[s_i(t_k)] = s_i(t_k) * \frac{1}{2\pi}\left[\int_{-\pi}^{0} j \cdot e^{jwk}dw - \int_{0}^{\pi} j \cdot e^{jwk}dw\right]$$

where $k = 1$ to T, $\mathbf{A}_i(t)$ represents the instantaneous amplitude, and $\Phi_i(t)$ represents the instantaneous phase. The instantaneous phase synchronous (IPS [97]), which measured the phase similarity at each timepoint, can be calculated by the following,

$$IPS(t) = 1 - \sin\left(\frac{|\Phi_1(t) - \Phi_2(t)|}{2}\right)$$

where the phase is in the unit of degree. IPS spans the range of 0–1, where a larger value indicates a stronger synchrony. We then define a quarter of the whole range of phase difference (180˚), 45˚, as the threshold. When the phase difference is less than 45˚, IPS was greater than 0.62, thus revealing a better performance. We finally calculated the PSI by the ratio of the time with the IPS greater than 0.62,

$$PSI = \frac{t_{IPS} > 0.62}{T}$$

## Supporting information

**S1 Fig. Benchmark with NetPyNE.** Scatter plots of MAE in the time domain, MAE in the frequency domain and PSI in the phase domain. Empty circles indicate overall average MAEs and PSI values for msDyNODE (black: firing rate, blue: LFP) and NetPyNE (red). Dim points represent average MAEs and PSI over trials for each recording channel. *p < 0.05, **p < 0.01, ***p < 0.001 using two-sided Wilcoxon's rank-sum test.
(TIF)

**S2 Fig. Benchmark with M-SGM and SBI-SGM.** Periodograms of MAEs in frequence responses spanning from 0 to 40 Hz.
(TIF)

**S3 Fig.** Granger causality-based graph properties over eight different target directions for Monkey A and B. Number of edges, average clustering, and number of total triangles derived from Granger causality-based excitatory (blue) and inhibitory (red) subnetworks are presented in polar coordinated for Monkey A (top) and B (bottom), respectively.
(TIF)

**S4 Fig. DTF-based graph properties over eight different target directions for Monkey A and B.** Number of edges, average clustering, and number of total triangles derived from Granger causality-based network are presented in polar coordinated for Monkey A (top) and B (bottom), respectively.
(TIF)

## Acknowledgments

We thank José del R. Millán from Clinical Neuroprosthetics and Brain Interaction lab at University of Texas at Austin for extensive discussion and suggestions.

## Author Contributions

**Conceptualization:** Yin-Jui Chang, Samantha R. Santacruz.

**Data curation:** Yin-Jui Chang, Hannah M. Stealey, Yi Zhao, Hung-Yun Lu, Enrique Contreras-Hernandez, Megan N. Baker, Samantha R. Santacruz.

**Formal analysis:** Yin-Jui Chang, Yuan-I Chen, Samantha R. Santacruz.

**Funding acquisition:** Hsin-Chih Yeh, Samantha R. Santacruz.

**Investigation:** Yin-Jui Chang, Hannah M. Stealey, Yi Zhao, Hung-Yun Lu, Enrique Contreras-Hernandez, Megan N. Baker.

**Methodology:** Yin-Jui Chang, Yuan-I Chen, Edward Castillo, Hsin-Chih Yeh, Samantha R. Santacruz.

**Project administration:** Hsin-Chih Yeh, Samantha R. Santacruz.

**Resources:** Samantha R. Santacruz.

**Software:** Yin-Jui Chang.

**Supervision:** Edward Castillo, Hsin-Chih Yeh, Samantha R. Santacruz.

**Validation:** Yin-Jui Chang.

**Visualization:** Yin-Jui Chang.

**Writing – original draft:** Yin-Jui Chang, Samantha R. Santacruz.

**Writing – review & editing:** Yin-Jui Chang, Yuan-I Chen, Hannah M. Stealey, Yi Zhao, Hung-Yun Lu, Enrique Contreras-Hernandez, Megan N. Baker, Edward Castillo, Hsin-Chih Yeh, Samantha R. Santacruz.

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
