## [Decision Letter · Decision Letter 0]

6 Aug 2024

PONE-D-24-13654Multiscale effective connectivity analysis of brain activity using neural ordinary differential equationsPLOS ONE

Dear Dr. Santacruz,

Thank you for submitting your manuscript to PLOS ONE. After careful consideration, we feel that it has merit but does not fully meet PLOS ONE’s publication criteria as it currently stands. Therefore, we invite you to submit a revised version of the manuscript that addresses the points raised during the review process.

We look forward to receiving your revised manuscript.

Kind regards,

Marko Čanađija

Academic Editor

PLOS ONE

Journal Requirements:

**Additional Editor Comments:**

Dear Prof. Santacruz,

since you have been waiting for the second review for quite some time and since two reviewers have accepted but not completed the task, I have decided not to wait any longer for the second review. Please address the reviewer's concerns carefully and submit the revised manuscript.

Marko Čanađija

Reviewers' comments:

Reviewer's Responses to Questions

**Comments to the Author**

1. Is the manuscript technically sound, and do the data support the conclusions?

Reviewer #1: Partly

2. Has the statistical analysis been performed appropriately and rigorously? 

Reviewer #1: Yes

3. Have the authors made all data underlying the findings in their manuscript fully available?

Reviewer #1: No

4. Is the manuscript presented in an intelligible fashion and written in standard English?

Reviewer #1: Yes

5. Review Comments to the Author

Reviewer #1: This article deals with the crucial question of the multiscale nature of brain functioning. The authors proposed a tool to estimate multiscale effective connectivity via neural ordinary differential equations to model multiscale dynamics. The authors validated their model both via simulations and experimental data.

The paper is well-written and easy to follow.

The authors put a lot of efforts in testing their methods in different configurations (both in simulations and in real data). However, some crucial elements to demonstrate the power of their model are missing.

First, there is no benchmark with existing estimators of causal brain connectivity (such as Granger causality, Directed Transfer Function, or Dynamic Causal Modelling [1] for instance), that could be coupled with multiplex approaches [2] to study multiscale networks.

Secondly, there is no benchmark with existing biophysical models of brain functioning such as NetPyNE [3], or spectral graph theory-based models [4] and its Bayesian version [5]. In the latter case, the model links excitatory and inhibitory neuronal responses to the oscillatory activity that was a target identified by the authors in the manuscript.

In that spirit, additional assessments (beyond the MAE) should be proposed to better appreciate the performance of the proposed model. An evaluation of the classification performance associated to the prediction of the ground truth in the case of real data, and an assessment of the computational time to complete the analysis.

Finally, further information on the deep learning architecture that has been chosen would be important to understand all the steps that are required to build the model.

Minor comment:

Could the author update the GitHub repository please? With the current version of the files, it is not possible to load the data for the toy example

References

[1] O. David, S. J. Kiebel, L. M. Harrison, J. Mattout, J. M. Kilner, et K. J. Friston, « Dynamic causal modeling of evoked responses in EEG and MEG », NeuroImage, vol. 30, no 4, p. 1255‑1272, mai 2006, doi: 10.1016/j.neuroimage.2005.10.045.

[2] M. De Domenico, « Multilayer modeling and analysis of human brain networks », GigaScience, vol. 6, no 5, p. 1‑8, févr. 2017, doi: 10.1093/gigascience/gix004.

[3] S. Dura-Bernal et al., « NetPyNE, a tool for data-driven multiscale modeling of brain circuits », eLife, vol. 8, p. e44494, avr. 2019, doi: 10.7554/eLife.44494.

[4] P. Verma, S. Nagarajan, et A. Raj, « Spectral graph theory of brain oscillations—-Revisited and improved », NeuroImage, vol. 249, p. 118919, avr. 2022, doi: 10.1016/j.neuroimage.2022.118919.

[5] H. Jin, P. Verma, F. Jiang, S. S. Nagarajan, et A. Raj, « Bayesian inference of a spectral graph model for brain oscillations », NeuroImage, vol. 279, p. 120278, oct. 2023, doi: 10.1016/j.neuroimage.2023.120278.

6. PLOS authors have the option to publish the peer review history of their article (what does this mean?). If published, this will include your full peer review and any attached files.

Reviewer #1: No

---

## [Author Response · Author response to Decision Letter 0]

21 Oct 2024

Dear reviewers,

We thank the reviewers for their valuable comments which have enabled us to develop this substantially stronger revised manuscript. The revision has been edited and reviewed in consultation with all contributing authors, and each author has approved the final form of this revision. The following is our point-by-point response to the reviewers’ comments.

Response to Reviewer 1:

 This article deals with the crucial question of the multiscale nature of brain functioning. The authors proposed a tool to estimate multiscale effective connectivity via neural ordinary differential equations to model multiscale dynamics. The authors validated their model both via simulations and experimental data. The paper is well-written and easy to follow. The authors put a lot of efforts in testing their methods in different configurations (both in simulations and in real data). However, some crucial elements to demonstrate the power of their model are missing.

Thank you for carefully reading our manuscript. We appreciate the reviewer’s valuable advice that allowed us to improve our work, especially by providing more comparisons with existing methods and more metrics to demonstrate the power of our approach. Please find below our point-by-point response to your comments.

 First, there is no benchmark with existing estimators of causal brain connectivity (such as Granger causality, Directed Transfer Function, or Dynamic Causal Modelling [1] for instance), that could be coupled with multiplex approaches [2] to study multiscale networks.

We thank the reviewer for this comment. In dynamical causal modeling, the brain is treated as a nonlinear dynamical system to determine the influence one neuronal system exerts over another [1]. By definition, our approach is also categorized as dynamical causal modeling. The difference between our approach and the reference indicated in the review is the method to obtain the optimal parameters, where the former utilizes deep learning techniques, and the latter uses Bayesian inference. Therefore, we posit that the dynamic causal modeling described in [1] is not an appropriate benchmark. However, the reviewer has raised an important point regarding benchmarking our method with existing estimators causal brain connectivity. In the revised manuscript, we include new comparisons (see Figs. S3-S4) with the existing estimators of causal brain connectivity, including Granger causality [2] and Directed Transfer Function (DTF; [3]). Granger causality is quantified by how well a time series signal can predict another time series signal. The typical model utilized is the vector autoregression (VAR) model. Since our approach only includes one lag, the specific model implemented here for estimating Granger causality is a first-order VAR model. Then the Granger causality is determined by the log-ratio between the error variance of a reduced model and that of the full model. On the other hand, DTF is determined based on the spectral form of the multichannel autoregressive model (MVAR) with n channels, where we can obtain the transfer matrix of the system, H. Similar to VAR in Granger causality estimation, the order of MVAR is set to be 1. Accordingly, the DTF from i to j at frequency f, DTFij(f), is determined by the following,

DTF_ij (f)=|H_ij (f)| (∑_(k=1)^n▒|H_ik (f)|^2 )^(-1/2) 

We apply both measures on both monkey datasets, quantify the connectivity using the common graph properties such as the number of edges, average clustering, and total triangles, and summarize them in Figures R1-2 below. In Figure R1, Granger causality indicated different graph properties across the eight target directions, supporting our conclusion that both excitatory and inhibitory multiscale effective connectivity (msEC) exhibit unique patterns relating to target directions (Figure 6c). However, the DTF is unable to be divided into excitatory and inhibitory subnetworks. Therefore, the differences in DTF-based graph properties are relatively small across target directions (Figure R2), suggesting that the unique patterns in excitatory and inhibitory subnetworks cannot be observed in such an ensemble measure. 

Taken together, although Granger causality and DTF are powerful tools for characterizing functional coupling between brain regions, they primarily reflect the patterns of statistical dependence. To better reveal the causal interactions that align with the actual mechanisms of brain function, it is suggested to assess effective connectivity using a mechanistic model, such as the msEC proposed in our study. We add the summarized benchmark with existing estimators of causal brain connectivity in the Discussion section of the revised manuscript and the Supporting Information. 

Page 7, line 11-18 (Discussion): Although existing estimators of causal brain connectivity (e.g., Granger causality [82] and directed transfer function (DTF [83])) provide disparate graph properties (Figs. S3-4), Granger causality supports our observation that both excitatory and inhibitory msEC exhibit unique patterns relating to target directions. In contrast, DTF failed to demonstrate unique patterns, which may be due to its inability to be divided into excitatory and inhibitory subnetworks. While both existing estimators are powerful tools for characterizing functional coupling between brain regions, they primarily reflect the patterns of statistical dependence. To better reveal the causal interactions that align with the actual mechanisms of brain function, it is suggested to assess effective connectivity using a mechanistic model, such as msDyNODE.

3. Secondly, there is no benchmark with existing biophysical models of brain functioning such as NetPyNE [3], or spectral graph theory-based models [4] and its Bayesian version [5]. In the latter case, the model links excitatory and inhibitory neuronal responses to the oscillatory activity that was a target identified by the authors in the manuscript.

We thank the reviewer for raising the issue of missing benchmarks with existing biophysical models of brain functioning. To demonstrate the power of our proposed model, we apply three existing approaches: NetPyNE [4], modified spectral graph theory model (M-SGM [5]), and SGM integrated with simulation-based inference for Bayesian inference (SBI-SGM [6]), on ten randomly selected trials of Monkey A dataset and compare these model performances (Figures R3-4). 

NetPyNE is a powerful tool to develop data-driven multiscale network models. In the network, corticostriatal neurons are considered to be excitatory populations (a total of 150 cells), and fast-spiking neurons are included for inhibitory populations (a total of 150 cells). In the simulation configuration, “LFP recording” is enabled to generate the LFP time series. To obtain the firing rate at the same LFP channel, we assign each cell to the specific LFP channel based on the distance and calculate the firing rate per channel from the spikes from the assigned cells. We perform a grid parameter search by setting up a batch simulation in NetPyNE to find the optimal parameters that best fit the real data. As demonstrated in the revised manuscript, we compared the performance of our approach, msDyNODE, with NetPyNE in the time domain, the frequency domain, and the phase domain (Figure R3). The msDyNODE is superior to NetPyNE by showing smaller MAEs in both time and frequency domains, and a greater phase synchronization with the ground truths. The potential reason for relatively poor performance in NetPyNE may be due to inaccurate modeling. Indeed, NetPyNE is a powerful tool to define the model at molecular, cellular, and circuit scales when the model parameters such as populations, cell properties, and connectivity are accurate. Although NetPyNE also provides evolutionary algorithms beyond the grid parameter search to perform parameter optimization and exploration, the improper selection of parameters and their ranges to be optimized can degrade the performance. 

M-SGM provides a closed-form solution of brain oscillations in the form of steady-state frequency response using the eigendecomposition of the structural connectome’s Laplacian [7]. A dual annealing optimization [8] is performed to estimate the optimized parameters. SBI-SGM is further proposed to infer the posterior distribution of the SGM parameters using simulation-based inference [9]. Since we do not have diffusion MRI data for structural connectivity networks, we make these parameters optimizable in the implementation. Due to the fact that SGM yields frequency responses, we compared MAE in the power spectrum with the real data for msDyNODE, M-SGM, and SBI-SGM (Figure R4), demonstrating that msDyNODE exhibits better performance than SGMs. msDyNODE models the real multiscale brain signals better than SGMs may be due to the consideration of nonlinear brain dynamics and spatially varying parameters. 

Another advantage of msDyNODE over NetPyNE, M-SGM, and SBI-SGM is the adaptability of the new model. For msDyNODE, the user can easily modify the differential equation sections in the script. However, NetPyNE requires the development of an external module in NEURON [10,11]. For M-SGM and SBI-SGM, the new transfer function is required to derive from the new models. 

In summary, the proposed approach in this study, msDyNODE, is demonstrated superior to existing biophysical models of brain functioning, including NetPyNE, M-SGM, and SBI-SGM, by showing better performance and the feasibility of customizing user needs. We add the summarized benchmark with existing biophysical models of brain functioning in the Discussion section of the revised manuscript and the Supporting Information.

Page 6, line 22-35 (Discussion): Comparing with existing biophysical models of brain functioning, including NetPyNE [61], modified spectral graph theory model (M-SGM [62]), and SGM integrated with simulation-based inference for Bayesian inference (SBI-SGM [63]), we demonstrate that msDyNODE is superior to these approaches. msDyNODE showed smaller MAEs in both time and frequency domains, and a greater phase synchronization with the ground truth signals (Fig. S1). The potential reason for relatively poor performance in NetPyNE may be due to inaccurate modeling. Indeed, NetPyNE is a powerful tool to define the model at molecular, cellular, and circuit scales when the model parameters such as populations, cell properties, and connectivity are accurate. Although NetPyNE also provides evolutionary algorithms beyond the grid parameter search to perform parameter optimization and exploration, the improper selection of parameters and their ranges to be optimized can degrade the performance. Furthermore, msDyNODE exhibits better performance than both versions of SGMs (Fig. S2). The rationale for why msDyNODE models the real multiscale brain signals better than SGMs may be due to the consideration of nonlinear brain dynamics and spatially varying parameters. Another advantage of msDyNODE over NetPyNE, M-SGM, and SBI-SGM is the adaptability of the new model. For msDyNODE, the user can easily modify the differential equation sections in the script. However, NetPyNE requires the development of an external module in NEURON [64,65]. For M-SGM and SBI-SGM, the new transfer function is required to derive from the new or customized models.

 In that spirit, additional assessments (beyond the MAE) should be proposed to better appreciate the performance of the proposed model. An evaluation of the classification performance associated to the prediction of the ground truth in the case of real data, and an assessment of the computational time to complete the analysis.

We appreciate the reviewer’s point on providing additional assessment beyond MAE. Given good performances in the time domain based on MAE, it is expected that the proposed model maintains the integrity of information represented by the neural activity, and thus the classifier trained with the real data and that trained with the model prediction can yield the comparable decoding capability. Evaluation of the classification accuracy on real data and model predictions may not add value to the performance of the proposed model. Instead, we provide two additional assessments on different domains: frequency and phase. In frequency domain, we assess the model performance in the frequency domain by calculating the MAE of the power spectrum. In phase domain, we evaluate whether the predicted and the real signals are phase-synchronized by calculating the phase synchrony index (PSI [12]). We first utilized the Hilbert transform, HT[·], to obtain the instantaneous phase of two time series signals, Φ1(t) and Φ2(t), from the pair of signals, s1(t) and s2(t).

z_i (t)=s_i (t)+j HT[s_i (t)]=A_i (t) e^(jϕ_i (t))

HT[s_i (t_k )]=s_i (t_k )*1/2π [∫_(-π)^0▒〖j∙e^jwk dw〗-∫_0^π▒〖j∙e^jwk dw〗]

where k = 1 to T, Ai(t) represents the instantaneous amplitude, and Φi(t) represents the instantaneous phase. The instantaneous phase synchronous (IPS), which measured the phase similarity at each timepoint, can be calculated by the following,

IPS(t)=1-sin⁡(|ϕ_1 (t)-ϕ_2 (t)|/2)

where the phase is in the unit of degree. IPS spans the range of 0-1, where a larger value indicates a stronger synchrony. We then define a quarter of the whole range of phase difference (180°), 45°, as the threshold. When the phase difference is less than 45°, IPS was greater than 0.62, thus revealing a better performance. We finally calculated the PSI by the ratio of the time with the IPS greater than 0.62,

PSI=(t_IPS>0.62)/T

These two measures are both applied to both monkey datasets, and the results are summarized in Figure R5. Overall, the msDyNODE well captures the signal’s power across 0 to 40 Hz for both Monkey A (Figure R5a-b) and Monkey B (Figure R5d-e). In addition, msDyNODE-predictions are in sync with the ground truths by showing most of the predictions with PSI greater than 0.5 (Figure R5c,f). The model performance in the frequency and phase domains align well with that in the time domain, where the msDyNODE predicts LFP better with more channels included in the model. 

Despite accurate modeling of multiscale neural activity, the major limitation of the proposed approach is the network training, which is computationally expensive and has long training times (on the order of hours). In contrast, the prediction process is significantly faster, which takes several minutes to complete the prediction. To mitigate the issue on long network training time, transfer learning [13] from a previously trained network can potentially speed up the convergence of the learning process. In the future work, we will investigate how transfer learning techniques can improve the computation time to complete the analysis.

In summary, we have included additional assessments on the performance of the model, including the predictions in the frequency and phase domains and the computational time, in the Results, Discussion, and Materials and Methods sections. Figure R5 is integrated into original Figs. 4 and 5 in the revised manuscript.

Page 4, line 20-25 (Results): Beyond MAE in the time domain, we also assess MAE in the frequency domain and phase synchronization in the phase domain (Fig. 4f-h, Fig. 5f-h; see Materials and Methods). Overall, the msDyNODE captures the signal’s power for both Monkey A (Fig. 4f,g) and Monkey B (Fig. 5f,g). Notably, phase synchronization is recognized as a fundamental neural mechanism that supports neural communication and plasticity [40]. Therefore, the model performance in the phase domain is crucial. We demonstrated that msDyNODE-predictions are in sync with ground truth by showing most of the predictions have a phase synchrony index greater than 0.5 (Fig. 4h, Fig. 5h).

Page 7, line 3-6 (Discussion): Furthermore, hours of network training time are another issue for quick implementation. In future work, transfer learning [76] from the previously trained network may be a possible strategy to improve computation time by potentially speeding up the convergence of the learning process.

5. Finally, further information on the deep learning architecture that has been chosen 

---

## [Decision Letter · Decision Letter 1]

8 Nov 2024

Multiscale effective connectivity analysis of brain activity using neural ordinary differential equations

PONE-D-24-13654R1

Dear Dr. Santacruz,

We’re pleased to inform you that your manuscript has been judged scientifically suitable for publication and will be formally accepted for publication once it meets all outstanding technical requirements.

Kind regards,

Marko Čanađija

Academic Editor

PLOS ONE

Additional Editor Comments (optional):

Reviewers' comments:

Reviewer's Responses to Questions

**Comments to the Author**

1. If the authors have adequately addressed your comments raised in a previous round of review and you feel that this manuscript is now acceptable for publication, you may indicate that here to bypass the “Comments to the Author” section, enter your conflict of interest statement in the “Confidential to Editor” section, and submit your "Accept" recommendation.

Reviewer #1: All comments have been addressed

2. Is the manuscript technically sound, and do the data support the conclusions?

Reviewer #1: Yes

3. Has the statistical analysis been performed appropriately and rigorously? 

Reviewer #1: Yes

4. Have the authors made all data underlying the findings in their manuscript fully available?

Reviewer #1: Yes

5. Is the manuscript presented in an intelligible fashion and written in standard English?

Reviewer #1: Yes

6. Review Comments to the Author

Reviewer #1: (No Response)

7. PLOS authors have the option to publish the peer review history of their article (what does this mean?). If published, this will include your full peer review and any attached files.

Reviewer #1: No

---

## [Editor Report · Acceptance letter]

22 Nov 2024

PONE-D-24-13654R1 

PLOS ONE

Dear Dr. Santacruz, 

I'm pleased to inform you that your manuscript has been deemed suitable for publication in PLOS ONE. Congratulations! Your manuscript is now being handed over to our production team.

Kind regards, 

on behalf of

Dr. Marko Čanađija 

Academic Editor

PLOS ONE